# Targets for CAR Therapy in Multiple Myeloma

**DOI:** 10.3390/ijms26136051

**Published:** 2025-06-24

**Authors:** Olga A. Bezborodova, Galina V. Trunova, Elena R. Nemtsova, Varvara A. Khokhlova, Julia B. Venediktova, Natalia B. Morozova, Maria S. Vorontsova, Anna D. Plyutinskaya, Elena P. Zharova, Peter V. Shegai, Andrey D. Kaprin

**Affiliations:** Moscow Hertsen Research Institute of Oncology—Branch of the National Medical Radiology Research Center, Ministry of Health of the Russian Federation, Moscow 125284, Russia; gtrunovamnioi@mail.ru (G.V.T.); nemtz@yandex.ru (E.R.N.); nostocus@yandex.ru (V.A.K.); uluaz@yandex.ru (J.B.V.); n.b.morozova@yandex.ru (N.B.M.); vapireshouse@mail.ru (M.S.V.); nestor2031@yandex.ru (A.D.P.); zharchik85@yandex.ru (E.P.Z.); dr.shegai@mail.ru (P.V.S.); kaprin@mail.ru (A.D.K.)

**Keywords:** multiple myeloma, CAR-T, CAR-NK, tumor targets, immunotherapy, cellular therapy, antigen escape, multi-specific CAR, logical activation, safety of therapy

## Abstract

Multiple myeloma (MM or plasma cell myeloma) is a heterogenous B-cell malignant tumor that typically exhibits a high recurrence rate, resistance to drugs, and molecular diversity of tumor subclones. Given the limited efficacy of standard therapy options, cellular immunotherapy featuring a chimeric antigen receptor (CAR) has proven tangible potential in treatment for relapsed and refractory forms of MM. The rational choice of a tumor target which shows high selectivity, stable expression, and biological significance is key to the successful implementation of CAR therapy. This review has summarized and analyzed data from the literature on biological properties, the features of expression, and the clinical development stages of CAR cell products for MM treatment which target BCMA, GPRC5D, FcRH5, SLAMF7, CD38, CD138, TACI, APRIL, CD19, TNFR2, CD44v6, CD70, NKG2D ligands, etc. Special focus is on strategic approaches to overcoming antigenic escape, such as multi-specific CAR constructs, logical activation sequences, and controlled safety systems. The analysis underscores the need for integrating the molecular selection of targets with cutting-edge bioengineering solutions as a key trend for raising the efficacy, stability, and safety of cellular therapy in the case of MM.

## 1. Introduction

Multiple myeloma (MM) is the second most widespread blood cancer disease and exhibits pronounced molecular heterogeneity, aggressive clinical progression, and a high risk of drug resistance occurrence [1,2].

Contemporary MM treatment regimens include the use of target drugs, such as primarily proteasome inhibitors, which have proven their high efficacy across different stages of the disease [3,4,5]. Nevertheless, the clinical application of such agents is limited by the development of drug resistance, including cross-resistance to other anti-tumor agents [6].

The utilization of immunotherapy with monoclonal antibodies, such as daratumumab and elotuzumab that are embedded in contemporary MM treatment standards and have become an integral part of combined therapy regimens, has evolved into an extra therapeutic area [7]. When combined with other drugs, this contributes to higher efficacy and partial overcoming of resistance but fails to solve the problem, i.e., completely eradicate the disease [8].

CAR (Chimeric Antigen Receptor)-based cellular therapy stands out as a breakthrough and promising approach to the treatment of relapsed and refractory MM. CAR T-cell therapy has shown a high frequency of therapeutic responses in patients that have used up the opportunities offered by standard treatment methods, which corroborates the tangible potential of the technology [9,10].

The B-cell maturation antigen (BCMA), which is expressed essentially on the surface of malignant plasma cells, appears as the best-known target for CAR therapy in the case of MM. However, the long-term efficacy of BCMA-targeted therapy is limited with the development of antigen-negative relapses and the depletion of effector T cells, which emphasizes the need for putting in place improved strategies and novel approaches to CAR creation [11,12].

This review is aimed to summarize and analyze data from the literature on biological properties, the features of expression, and the clinical stages of CAR cell products for MM treatment (Figure 1).

## 2. Structural Organization and Biological Fundamentals of CAR Cell Products in Immunotherapy of Blood Cancer Diseases

### 2.1. Background for CAR Cell Products in Therapy of MM

CAR cell product-based immunotherapy holds a key position among contemporary strategies for the treatment of blood cancer diseases such as MM due to its high specificity, capability for the targeted destruction of tumor cells, and efficacy in patients with resistant forms of the disease [13,14,15]. Central to this therapy are the chimeric antigen receptors (CARs), artificially constructed molecules that instill in T cells the capability for antigen-specific recognition, activation, and proliferation regardless of the major histocompatibility complex (MHC) molecules [16]. This mechanism is pivotal in tumor evolution and immune escape, since malignant cells oftentimes lose the expression of MHC molecules, thereby avoiding being recognized with traditional T-cell receptors [17].

### 2.2. Structural Organization of CAR Cell Products and Their Interaction with Tumor Antigen

The CAR construct incorporates three functional segments: an extracellular antigen-binding domain, a transmembrane domain, and intracellular signaling domains [18]. As a rule, the typical extracellular domain is a single-chain variable fragment of the scFv antibody with high specificity for tumor antigens [19]. The transmembrane domain acts as an anchor and fixes the CAR on the T-cell surface, facilitating its stable expression and the origination of multimeric complexes [20]. Intracellular signaling domains ensure the activation of a T cell in response to an interaction with the target, determining the range of its functional activity, including proliferation, the secretion of cytokines, and cytotoxicity [21].

The most well-known cellular products are used in contemporary cancer immunotherapy—CAR T cells and CAR NK cells, which differ by their cellular origin and constructional features of chimeric receptors [22,23,24].

The typical construct of CAR T cells uses CD3ζ—a component of the T-cell receptor (TCR) signaling pathway that ensures the imitation of the physiological activation of T lymphocytes as the intracellular signaling domain [25]. The co-stimulating domains CD28 and/or 4-1BB (CD137) are included into the CAR construct to enhance the antigen-specific response. CD28 ensures powerful primary activation of T cells and facilitates their quick expansion; in turn, 4-1BB maintains long-term survival and resistance to depletion and promotes the formation of memory T cells [16,26].

Typical constructs of CAR NK cells include signaling modules that are inherent for an innate immune system link—DAP10 (DNAX-activating protein 10) and DAP12 (DNAX-activating protein 12), as well as the co-stimulating receptor 2B4 (CD244). These elements initiate endogenous signaling cascades that are specific for the activation of NK cells, thus ensuring the effective cytotoxic activity and secretion of pro-inflammatory cytokines [27,28]. Unlike CAR T cells, CAR NK cells have a more controllable activation profile, which mitigates the risk origination of hyperactivation and a cytokine storm, thereby preserving pronounced anti-tumor efficacy.

In most CAR products, such as CAR T cells and CAR NK cells, a single-chain variable fragment of the scFv antibody is used as an extracellular antigen-binding domain [29]. However, some constructs of CAR NK cells use innate immunity receptors, for instance NKG2D (natural killer group 2D) and DNAM-1 (DNAX accessory molecule-1), as an extracellular domain. NKG2D recognizes NKG2D ligands (in total eight); among them two stress-associated molecules, MICA and MICB (MHC class I polypeptide-related sequence A/B), are expressed by tumor cells. The expression level of these proteins on tumor cells is higher than in untransformed ones, which makes them selective targets [30].

The tumor antigen primary recognition process with a chimeric receptor (CAR) starts with the interaction of the extracellular domain scFv with a specific epitope of a ligand molecule on the tumor cell surface [31]. The structural and chemical complementarity of scFv to the target epitope ensures high binding specificity, enabling it to effectively distinguish tumor cells and minimize damage to healthy cells. The following act as the key features in this interaction: affinity, i.e., the binding strength of one scFv with the epitope; avidity, i.e., the aggregate strength of multiple interactions between the CAR and antigens; and the uniqueness of a paratope, i.e., the individual properties of the antigen-binding center of a scFv. Together, these parameters play a crucial role in the functional outcome of the interaction between the effector and tumor cells, determining the strength, duration, and efficacy of the immune response [32].

### 2.3. Sources for Producing CAR Cell Products

Off-the-shelf CAR T cells are predominantly produced from autologous or allogeneic (donor) T lymphocytes that undergo successive release, activation, gene modification (transduction) and expansion stages prior to infusion into a patient. In CAR NK therapy the range of sources for obtaining the cell material are sizably wider. Adult peripheral blood, umbilical cord blood, the NK-92 cell line, and induced pluripotent stem cells (iPSCs) capable of differentiation into NK cells can be used to obtain NK cells. A variety of sources paves a way towards the development of personalized and standardized CAR NK products, such as ready-to-use off-the-shelf medications, which drastically expands the clinical opportunities and enhances the scalability of the therapy [28,33,34,35].

### 2.4. Safety of CAR Cell Products

Despite the high clinical efficacy in treatment for blood cancer diseases, CAR T-cell therapy is oftentimes accompanied by the development of grave side effects—cytokine release syndrome (CRS) and immune effector cell-associated neurotoxicity syndrome (ICANS) [13,36].

Unlike CAR T cells, CAR NK cells show a more favorable safety profile, stipulated by more balanced natural mechanisms for regulating the cytotoxic activity of NK cells and their reduced capability for uncontrollable production of pro-inflammatory cytokines [27,30,37].

The state-of-the-art condition of CAR cell therapy is characterized by rapid growth and high clinical efficacy, which is facilitated by solid achievements in genetic engineering of chimeric antigen receptors. Most sweeping innovations are related to the structural optimization of CAR T cells and CAR NK cells to the choice of the most suitable antigen targets to improve the safety profile of the therapy while preserving the stable anti-tumor activity. Nonetheless, despite the successes, the selectivity and stability of CAR interactions with tumor targets is still the key restrictive factor, as it determines both the immediate clinical effect and its duration.

To that effect, further evolution of this field is impossible without comprehensive engineering and molecular solutions with a view to overcome the immune escape mechanisms, boosting the affinity of the interaction and functional resistance of CAR cells.

Optimizing the “CAR—tumor target” axis is evolving as the pivotal area for prospective research, with the potential to substantially boost the efficacy and reliability of cellular immunotherapy.

### 2.5. Bioengineering Platforms and State-of-the-Art Strategies for Multi-Specific CAR Therapy in Cancer Immunotherapy

The development of poly-specific constructs that can enable a single population of T cells to simultaneously recognize several key tumor antigens has emerged as a priority area in contemporary cell engineering in the field of CAR therapy [38,39,40]. Such a strategy notably mitigates the risk of antigen escape, i.e., one of the basic mechanisms for tumor resistance to immunotherapy [41,42]. Moreover, poly-specific CAR platforms unveil new clinical prospects, including the potential for implementing the concept of personalized and adaptive cellular therapy that targets the molecular profile of a tumor in a specific patient [43].

Modern biotechnology strategies of obtaining biospecificity in CAR therapy pursue the creation of constructs that can recognize a bunch of tumor targets at once. The development of tandem CARs (TanCARs) with two or more successively located scFv domains is one of such approaches. Such extracellular domain architecture enables tandem CARS to simultaneously recognize various tumor antigens to improve the receptor-to-ligand bond strength, accuracy, and stability of the therapeutic response. An alternative is ligand-based CARs, which rely on the use of natural ligands in the extracellular domains of the receptor. An example is APRIL-CAR, which uses APRIL (a proliferation-inducing ligand) and is capable of binding with several receptors expressed on tumor cells—namely, BCMA and TACI (transmembrane activator and CAML interactor). Such an approach enables APRIL-CAR to expand the range of tumor target recognition via multi-ligand interactions while preserving the physiological specificity [44].

The development of bicistronic vectors that code various CARs facilitates the expression of two robust receptors in one cell, which promotes more effective targeting of tumor cells with a heterogenous profile of antigen expression [45]. An alternative approach, i.e., co-transduction, in which one T cell is modified for the expression of two or more CARs, ensures expanded antigen coverage of targets and may boost the efficacy of therapy in tumors with combined expression of antigens [46,47].

CAR constructs with logical activation elements, such as AND, NOT, and OR gates, have been developed in recent years to enhance selectivity and mitigate the risk of off-tumor toxicity. The AND gate concept states that CAR T-cell activation only occurs during the simultaneous recognition of two tumor antigens: one receptor initiates the activation signal, while the other implements co-stimulation [48]. NOT gates, which are implemented with the aid of inhibiting CARs (iCARs), function under the principle of suppressing the effector activity of a T cell during its interaction with antigens expressing on healthy cells and thereby ensure an extra safety control level. DiCAR constructs are made of several signaling domains and reliably inhibit the uncontrollable activity of effector cells. OR-gated CAR T cells are used to solve the problem of tumor cell heterogeneity. These cells are able to target various antigens on different tumor cells to provide better killing coverage [49,50,51].

Overcoming the immunosuppressive tumor microenvironment (TME) is a key problem for preserving the functional activity of CAR T cells in vivo [52]. For this purpose, armored CAR T cells of the third and subsequent generations are being developed; these have the capability to express the pro-inflammatory cytokines IL-12, IL-15, and IL-18 that augment the anti-tumor immune response. Such cells can additionally be modified for the expression of the dominant-negative transforming growth factor beta receptor II (TGF-βRII). TGF-βRII, which is expressed in CAR T cells, blocks the suppressive impact of TGF-β on T lymphocytes [53,54,55,56].

The next stage in the development of cellular therapy technologies is the creation of a fourth-generation CAR—the TRUCK platform (T cells redirected for universal cytokine killing), which ensures the local secretion of pro-inflammatory cytokines in response to a CAR binding with the tumor antigen [57,58]. This approach promotes the augmentation of the anti-tumor immune response and precludes the formation of an immunosuppressive TME [59,60,61].

Fifth-generation CAR T cells are additionally provided with intracellular domains that can activate signaling pathways associated with pro-inflammatory cytokines. These are elements of the JAK-STAT (Janus kinase—signal transducer and activator of transcription) signaling cascade, which ensures stable and autonomous cell activation even under unfavorable conditions [62].

Various molecular safety ‘switches’ that enable us to control the activity of modified T cells are integrated into the CAR construct to enhance the safety and controllability of the cellular therapy [63]. For instance, hypoxically inducible CARs activated solely under reduced oxygen conditions, typical for the tumor tissue, have been created. This approach ensures localized activation of CAR T cells and mitigates the risk of damage to normal tissues [64,65].

Relapses of the malignant process remain a frequent even after the minimal residual disease (MRD) state is achieved. As a rule, their occurrence is related to immune escape mechanisms due to the loss of target antigen expression by tumor cells and trogocytosis—the process in which the antigen is transferred to the CAR T-cell membrane and becomes unreachable for further recognition [66,67]. Multi-antigen CAR constructs capable of simultaneously recognizing several tumor targets, which drastically mitigates the risk of relapse occurrence and increases the stable effect of therapy, are being developed to overcome such restrictions [43,68].

The protocols based on combined or successful infusions of CAR products with varying antigen specificity levels attain extra clinical significance [69]. The use of similar treatment regimens is particularly promising in patients with refractory and relapsing forms of lymphomas, in whom successful therapy targeted at an expanded range of antigens promotes an increased frequency of remissions [70]. This novel therapeutic approach reduces the likelihood of relapse by eliminating the tumor clones with varying antigen expression levels, thereby minimizing the likelihood of immune escape [71,72].

In spite of the remarkable progress in the engineering of CAR constructs, the rational choice of a tumor target remains key to clinical efficacy. This problem becomes particularly relevant in pronounced molecular and phenotypic heterogeneity that is inherent for MM. In such cases, the stability of antigen expression on tumor cells, as well as their absence on normal tissues, has a direct impact on both the efficacy and safety of the therapy.

## 3. Tumor Targets in CAR Therapy for Multiple Myeloma

The rational choice of a tumor target is among the crucial stages in the development and clinical implementation of CAR cell therapy. It pursues the maximal therapeutic effect while simultaneously mitigating the risk of unwanted immunological complications, such as systemic toxicity and autoimmune reactions [73].

The presence of a tumor antigen that is capable of ensuring highly specific and reproducible recognition of pathological cells with minimal or absent expression in healthy tissues is key to the successful application of CAR therapy. This approach is critical to prevent the development of on-target, off-tumor toxicity—a hazardous phenomenon, in which CAR cells attack the body’s normal cells that express a low level of the target antigen. This phenomenon remains one of the main restrictions for the application of immunotherapy in the case of malignant neoplasms, including MM.

MM is characterized by pronounced molecular and phenotypic heterogeneity, which suggests the availability of multiple subclonal populations of tumor cells that differ by the level and range of surface antigen expression. This factor substantially complicates the choice of the single universal therapeutic target that can ensure an effective and selective impact [42,74].

When choosing an optimal antigen in the contemporary stage of CAR cell technology development, apart from its selectivity and expression level, several factors also account for a bunch of other critical parameters, including the biological role of the antigen in tumor pathogenesis and progression, expression stability in all subpopulations of myeloma cells, a link to the development of drug resistance, and the likelihood of cross-reactivity with normal tissues [75]. Moreover, the translational potential of a chosen target should also be analyzed for the scalability of manufacturing processes and clinical applicability, including the standardization and commercialization capability [76,77].

A comprehensive examination of tumor target features, including an analysis of their biological function, expression level, and stability on myeloma cells and an evaluation of their biological significance, clinical potential, and possible application restrictions as part of the CAR cell therapy, is a field in the development of more effective and safer MM treatment strategies [78,79]. Particularly noteworthy in this connection is reviewing the most promising targets with high utilization potential in the construct of CARs in the case of MM (Table 1).

### 3.1. B-Cell Maturation Antigen (BCMA)

BCMA, also known as TNFRSF17 (tumor necrosis factor receptor superfamily member 17), is a single-chain transmembrane protein that belongs to the tumor necrosis factor receptor superfamily (TNFRSF). Its construct includes an extracellular cysteine-rich domain (CRD), a transmembrane segment, and a short cytoplasmic domain [95,96]. These features of the construct stipulate for BCMA’s high selectivity as a therapeutic target for CAR cell therapy of MM, since the extracellular CRD ensures stable and specific binding with the scFv fragment of a CAR [97,98]. Standing as an extra benefit is the absence of tangible homology of BCMA’s extracellular area with other receptors, which mitigates the risk of cross-reactivity and on-target, off-tumor toxicity [99,100].

BCMA is expressed stably and with high density on the surface of malignant plasma cells, whereby it is virtually absent in other tissues, save for normal plasma cells [101]. This makes BCMA an attractive target not only for CAR cell therapy but also for other immunotherapy strategies.

At the molecular level, BCMA is engaged in the survival regulation of plasma cells. The interaction with the APRIL or BAFF (B-cell activating factor) ligand activates the NF-κB (nuclear factor kappa-light-chain-enhancer of activated B cells), PI3K/AKT (phosphatidylinositol-3-kinase/protein kinase B), and MAPK/ERK (mitogen-activated protein kinase/extracellular signal-regulated kinase) signaling cascades, which promote the growth, survival, and resistance of myeloma cells to apoptosis [101,102].

Clinical data corroborate that BCMA’s high expression level correlates with the intensity of the anti-tumor response, the likelihood of achieving complete remission, and the duration of relapse-free and overall survival [96,103].

Clinical trials, including KarMMa (for idecabtagene vicleucel, ide-cel) and CARTITUDE-1 (for ciltacabtagene autoleucel, cilta-cel), have shown high efficacy of BCMA-specific CAR T cells in patients with relapsing and refractory MM [104,105]. Idecabtagene vicleucel (ide-cel, Abecma; Bristol Myers Squibb Company, Summit, NJ, USA) is the first registered BCMA-specific CAR T product. The KarMMa trial reached an objective response rate (ORR) of 73%, complete remission (CR) in 33% of patients, a median overall survival duration of 19.4 months, and survival without progression—8.8 months. Side effects included CRS in 84% of patients (grade > 3 in 5%) and ICANS in 18% (grade > 3 in 3%) [103].

Ciltacabtagene autoleucel (cilta-cel, Carvykti; Legend Biotech Corporation, Somerset, NJ, USA) is the second-generation product that contains two scFv domains that recognize various BCMA epitopes, which ensures high avidity and efficacy [106,107]. The CARTITUDE-1 trial reached an ORR of 97%, and strict complete remission was observed in 67% of patients, whereas the median response duration exceeded 24 months. CRS emerged in 95% of patients (grade ≥ 3 in 5%) and ICANS in 21% of them, with mainly low or average severity.

Despite the impressive clinical results, the long-term efficacy of BCMA-specific CAR T therapy remains limited. The development of antigen-negative relapses resulting from the selective elimination of BCMA-positive cells and the subsequent survival of clones that lost their expression of this antigen is a key factor that precludes its stability [67]. The occurrence of disease relapses is linked to the existence of the soluble BCMA form (sBCMA) that circulates in the blood and competes with the membrane-bound BCMA for the CAR, thereby decreasing the efficacy of therapy and potentiating its immune escape [108,109].

To mitigate the risk of therapeutic inefficacy caused by antigen escape, poly-specific CAR products that simultaneously target the BCMA and extra tumor antigens, such as GPRC5D, FcRH5, and others, are being developed. Similar platforms expand the range of target antigens and boost the stability of the clinical effect due to more effective targeting of tumor cell heterogenous clones [97,99]. The development of high-affinity scFv domains is in the pipeline. These can selectively recognize solely the membrane-bound form of BCMA (mBCMA), excluding the interaction with its soluble form (sBCMA). This increases the specificity and efficacy of CAR cell activation [98,100].

Combined therapeutic strategies that stipulate the co-application of BCMA-targeted CAR T cells with immune checkpoint inhibitors, such as PD-1, and γ-secretase inhibitors, have been embedded in clinical practice. The latter promote a decrease in the level of soluble BCMA (sBCMA) and higher expression of membrane BCMA (mBCMA) on the surface of tumor cells, thereby boosting the efficacy of CAR T therapy [110,111].

Another novel field of development is the creation of armored CAR cells, which are stable in an immunosuppressive TME and capable of secreting pro-inflammatory cytokines, enhancing their proliferation and functional activity [78,112].

Along with autologous platforms, also progressing has been the field of creating allogenous off-the-shelf CAR T cells or CAR NK cells that target BCMA and are produced from iPSCs or from donor materials. These technologies enable us to standardize production, increase the affordability of therapy, and reduce its cost while ensuring controllable engineering modification to prevent allogenous immune reactions [112,113].

Hence, BCMA is worthily acting as the centerpiece among cellular therapy targets in MM due to a fusion of high specificity, stable expression, and its notable functional role in disease pathogenesis. The variety of approaches for the modification of BCMA-targeted platforms corroborates its strategic significance in existing promising therapeutic strategies.

### 3.2. G Protein-Coupled Receptor Class C Group 5 Member D (GPRC5D)

GPRC5D is a transmembrane protein falling within the superfamily of G protein-coupled receptors (GPCRs). It possesses a compact extracellular N-end domain, which makes it an easy target for coupling with the scFv contained in CARs [105,114]. GPRC5D’s high expression stability in myeloma cells and its accessibility on the tumor surface facilitate quick and effective interactions with CAR cells, ensuring reliable activation of the effector response [115,116].

GPRC5D expression outside the tumor tissue is restricted and mostly displayed increased expression in the stratified flat squamous epithelium, which mitigates the risk of on-target, off-tumor toxicity development [116,117].

The clinical and preclinical trials corroborate the favorable safety profile of GPRC5D-specific CAR products. Dermal toxicity related to GPRC5D expression in the epithelium is primarily registered in the mild and reversible form [118]. CRS is observed in 75–80% of patients, is mainly of severity grade I–II, and is well controlled [119,120]. ICANS is registered in less than 10% of patients and primarily flows in the mild form [117,121]. Hence, the toxicity profile of GPRC5D-targeted CAR products is deemed more favorable than in some BCMA-specific CAR systems [122].

For the first time, clinical efficacy was confirmed in the OriCAR-017 (phase 1, POLARIS) and CARTITUDE-2 trials, where the patients with relapsing or refractory MM, who had earlier received BCMA-targeted therapy, reached an overall response rate (ORR) of 70–90% [120,121].

The prevention of antigen escape is still a key problem in treatment for MM characterized by high heterogeneity. Bi-specific CAR products that are simultaneously targeting GPRC5D and BCMA, both in CAR-T and in the CAR-NK format, have been developed in this connection. Similar platforms display improved clinical outcomes and resistance to incidences of relapses [123,124]. Moreover, poly-specific constructs with the inclusion of IL-15 (armored CAR) and allogenous off-the-shelf CAR NK products have been actively promoted, which ensures additional affordability of the therapy, cost reduction, and potential for wide clinical application [125,126].

Hence, GPRC5D holds a prominent place among promising targets for cellular therapy of multiple myeloma, namely in patients with resistivity or relapsing after BCMA-specific therapy. The ongoing development of GPRC5D-targeted bi-specific and allogenous platforms opens up new opportunities for effective and safe treatment for this patient group.

### 3.3. Fc Receptor-Homolog 5 (FcRH5, CD307)

FcRH5 is a transmembrane glycoprotein that falls within the family of FcR-like receptors. It is mostly expressed on mature B cells, plasma cells, and malignant transformed cells in MM [111,127]. FcRH5’s restricted expression in normal conditions and its high abundance on tumor plasma cells substantiate its potential as a cellular therapy target, particularly in patients with a relapsing or refractory course of disease [128].

The extracellular segment of FcRH5 includes nine Ig-like domains, which represents a lengthy structure with multiple potential epitopes for coupling with CARs. Such architecture promotes high specificity and mitigates the risk of cross-reactivity [128]. The transmembrane domain of FcRH5 ensures solid fixation in the membrane and maintains stable expression on the surface of tumor cells—a critical condition for the stable effect of CAR therapy.

FcRH5 expression has been discovered in all subclones of myeloma cells, including those that originate after the loss of BCMA expression or resistance to BCMA-specific therapy [129]. This enables us to use FcRH5 as a target for second-line therapy and as a component for poly- and bi-specific platforms [130]. A series of preclinical trials has indicated that FcRH5-specific CAR T cells possess high cytotoxicity in in vitro and in vivo models, which results in the complete eradication of tumor cells without substantial toxic effects [111,128].

Also, FcRH5-targeted CAR NK platforms have been actively developed, and in preclinical models they demonstrate high specificity, a low alloreactivity risk, and a potential for large-scale off-the-shelf production [131]. The clinical trials of FcRH5-specific CAR T and FcRH5-targeted bi-specific antibodies show predictable and controllable toxicity profiles. The major side effects include CRS and ICANS, which are generally reversible and of moderate severity [132,133].

Due to its high expression and antigen stability, FcRH5 has been increasingly included into poly- and bi-specific CAR platforms (for instance, FcRH5+BCMA and FcRH5+GPRC5D), which is particularly promising for combating antigen escape in the case of tumor heterogeneity [39,134].

As indicated by preclinical trials, the inclusion of the co-stimulating domain 4-1BB (CD137) or ICOS (inducible t-cell costimulator) into FcRH5-targeted CAR constructs substantially improves the proliferative activity, persistence, and functional stability of effector cells in vivo. This ensures a steady and prolonged clinical effect, which is particularly crucial in refractory forms of MM [135,136,137,138].

In phase I–II clinical trials, FcRH5-specific CAR T cells and bi-specific antibodies have demonstrated a high overall response rate even in patients who had earlier been resistant to BCMA-targeted drugs [129]. Mostly moderate side effects have been revealed, which confirms the favorable profile of manageable toxicity.

Thus, FcRH5 has been viewed as the most promising and clinically meaningful target in the arsenal of cellular therapy for multiple myeloma. Its application as a target for new generations of CAR products enables us to expand the range of therapy, enhance resistance to relapsing, and provide treatment for patients who do not respond to BCMA-targeted strategies.

### 3.4. CD38 as a Therapeutic Target for Cellular Immunotherapy

CD38 is a type II transmembrane glycoprotein with a wide array of biological functions. It is engaged in intercellular adhesion and acts as a component in cellular signaling systems [139]. Owing to its enzymatic activity, CD38 is engaged in the regulation of metabolism, energy metabolism, and the innate and adaptive immune responses [96,140].

The molecular structure of CD38 includes a short cytoplasmic N-end fragment, a transmembrane domain, and a lengthy extracellular segment that is responsible for the catalysis of enzymatic reactions and the interaction with the extracellular matrix components [141].

CD38 has high and stable expression on the surface of myeloma plasma cells in all the disease stages, which makes it an attractive target for targeted therapy [142,143]. This marker is also expressed in a number of immune system normal cells (activated T and B lymphocytes, NK-cells, and monocytes) and in epithelial cells (for instance, type II alveolocytes) and lymphoid organs, which boosts the risk for the incidence of on-target, off-tumor toxicity.

CAR constructs that are able to recognize cells with high expression of CD38 typical for myeloma have been developed with a view to increase selectivity and decrease toxicity [65,144]. Likewise constructs with controllable inactivation of CAR cells (safety switches), have been developed—for instance, inducible apoptotic cascades (iCasp9) or pharmacologically controlled suppression of CAR signaling [145,146].

Several preclinical trials have displayed the high cytotoxic activity of CD38-specific CAR T cells both for in vitro and in vivo models [146]. Results of phase I clinical trials (for example, NCT03473496) have confirmed the relative safety and efficacy of CD38-specific CAR T cells in patients with relapsing and refractory multiple myeloma [96,147,148].

The creation of bi-specific CAR constructs, which can simultaneously recognize CD38 and BCMA, is a promising field. Similar platforms show a synergistic effect, boosting efficacy and decreasing the likelihood of resistance occurrence [149,150].

Altogether, the further optimization of CARs towards higher efficacy and safety and the adoption of poly-specific approaches unveil new opportunities for expanding the therapeutic window and clinical applicability of CD38-specific products in multiple myeloma.

### 3.5. Signaling Lymphocytic Activation Molecule Family 7 (SLAMF7)

SLAMF7/CD319 is a transmembrane glycoprotein from the family of SLAM receptors, playing a crucial role in the coordination of the innate and adaptive immune response [151]. The expression of SLAMF7 has been mainly observed on plasma cells, NK cells, dendritic cells, and individual subpopulations of T lymphocytes while nearly absent on hematopoietic stem cells, which makes the molecule an attractive target for cellular immunotherapy [152].

The SLAMF7 construct includes two extracellular immunoglobulin-like domains—a distal V domain and a proximal C2 domain, both engaged in intermolecular interactions and recognition by effector cells [153,154]. As indicated by preclinical trials, both domains may serve as independent epitopes for the creation of CAR constructs with varying functional profiles. However, the high expression of SLAMF7 on NK cells creates an autocytotoxicity risk from SLAMF7-targeted CAR T cells, which may weaken the natural immune surveillance and increase the likelihood of infectious complications [9,155,156].

To minimize the risk, various modifications of both CAR T and CAR NK platforms have been implemented using genome editing technologies and the introduction of inhibiting signaling domains that enable the preservation of the anti-tumor activity [157].

The efficacy and high specificity of SLAMF7-specific CAR T cells has been corroborated by both preclinical models and in early phases of clinical trials. In a phase I trial (NCT03710421), the overall response rate (ORR) exceeded 50%, including cases of complete and strict complete remission. CRS was mainly observed in the mild or moderate form, without multiple episodes of heavy neurotoxicity [147,158,159].

Alongside CAR-T, SLAMF7-targeted CAR NK-cells, which represent a promising platform due to reduced toxicity and the capability of delivering off-the-shelf products, have been actively developed.

Hence, the SLAMF7 glycoprotein is a highly specific, biologically substantiated, and clinically proven target for cellular therapy in multiple myeloma. SLAMF7 inclusion into poly-specific CAR constructs that can simultaneously recognize several antigens may drastically boost anti-tumor efficacy and mitigate the risk of antigen escape.

### 3.6. CD138 (Syndecan-1) as a Therapeutic Target for Cellular Immunotherapy

CD138 (syndecan-1) is a transmembrane heparan sulfate proteoglycan falling within the syndecan family. It plays a key role in the regulation of cellular adhesion and migration, transferring signals inside the cell in coupling with growth factors, morphogens, chemokines, enzymes, and extracellular matrix components [160].

The extracellular domain of CD138 contains heparan and chondroitin sulfate chains that ensure an interaction with extracellular matrix components. The transmembrane component fixes a molecule in the membrane, whereas a highly restricted short cytoplasmic section is engaged in signal transfer via the cytoskeleton. Such a structure enables the engagement of CD138 in a wide range of physiological and pathological processes, including tumor growth and invasion. The high accessibility of the extracellular section makes CD138 an easy target for coupling with CARs and the subsequent activation of T-cell effector functions [151].

CD138 is distinct for its stable and intense expression on MM cells [161,162]. Normally, protein is expressed on mature plasma and epithelial cells, which is used in diagnostics, whereas its expression on hematopoietic stem cells is minimal, which mitigates the risk of damage to the hematological system’s regenerative pool [163,164]. However, CD138 presence on the epithelium of various organs may stimulate the growth of off-target toxicity when using CD138-specific CAR T-cells.

Engineering strategies are being adopted to minimize such effects: lower affinity CARs are used, as they are capable of differentiating the tumor and normal cells by antigen expression level [144], as well as systems for the controllable shutdown of CAR cells (safety switches) in the development of undesired toxicity.

Preclinical trials have identified high-cytotoxicity Cd138-targeted CAR T cells, both in vitro and in vivo [161]. Clinical data, including phase I testing (NCT03778346), have confirmed the acceptable safety profile: As a rule, CRS was mild; neurotoxicity was rare, and the response level varied from partial to complete and strict complete remissions [164]. Moderate persistence of CAR cells was observed, which emphasizes the need for the optimization of their viability and functional robustness [162].

Thus, CD138 remains a meaningful target for cellular therapy in multiple myeloma. Despite potential restrictions, further evolution of CAR engineering and streamlining of effector cell activation control open up the prospects for enhancing the efficacy and clinical applicability of CD138-specific approaches.

### 3.7. CD70 as a Therapeutic Target for Cellular Immunotherapy

CD70 is a transmembrane protein from the TNF receptor family. It is expressed primarily on activated B and T lymphocytes, on dendritic cells, and on a series of malignant transformed cells, including plasma cells in MM [165].

The unique molecular characteristics of CD70 stipulate for its high selectivity as a target for cellular therapy. Namely, CAR constructs that are based on the CD70 interaction with its natural ligand CD27 demonstrate high rates of expansion and persistence of effector cells in vivo, which are required to maintain the steady anti-tumor effect [166].

Preclinical trials have confirmed the high cytotoxic activity of CD70-specific CAR T cells with respect to myeloma cells, whereby the low toxicity of normal tissues has been noted, including hematopoietic stem cells, thanks to minimal CD70 expression of extratumor or activated immune cells [167,168]. Further optimization of CAR constructs via molecular modeling of the CD70-CD27 interaction has enabled us to enhance their proliferation, survival, and specificity in the action of modified T cells.

Allogenous CD70-specific CAR T products that are constructed using CRISPR/Cas9 technology have been actively developed in recent years. They pass clinical evaluations in patients with refractory MM forms and solid tumors (CTX131 and NCT04554425 trials), demonstrating the acceptable safety profile in early phases [169,170].

The development of CD70-targeted CAR NK cells has been underway. Preclinical trials have demonstrated their high specificity and efficacy. The first clinical data corroborate the potential of this approach for MM therapy, especially in the case of CD19-negative forms of the disease [171,172,173].

Dual-targeting strategies that are simultaneously aimed at CD70 and BCMA and aid to overcome clonal heterogeneity and minimize the risk of antigen escape have been particularly interesting [174].

Hence, CD70 is a highly specific and biologically substantiated target for cellular therapy in multiple myeloma. The integration of CD70-targeted CAR T and CAR NK platforms, including bi-specific constructs, opens up prospects for increasing therapeutic efficacy, deepening and prolonging remission in patients with relapsing and refractory MM forms.

### 3.8. NKG2D Ligands (NKG2DLs) as a Target for Cellular Therapy of Multiple Myeloma

NKG2D ligands are stressed-induced molecules that include MICA, MICB, and the ULBP protein family (UL16-binding proteins) [175]. These molecules are expressed on the surface of cells in response to stresses, such as DNA damage, hypoxia, viral infection, and tumor transformation [176]. With respect to expression in normal tissues, the level of NKG2D ligands remains very low. Such selectivity makes the molecules promising targets for immunotherapy of multiple myeloma (MM).

The NKG2D receptor is expressed on NK cells, γδ-T cells, and CD8^+^ cytotoxic T lymphocytes, enabling the interaction between innate and adaptive immunity [177]. The binding of NKG2D ligands on the surface of tumor cells with the receptor on effector immune cells triggers powerful immune activation and destruction of tumor cells even in the case of phenotypic heterogeneity of the tumor [178]. This mechanism helps overcome antigen escape, attack various tumor clones, and maintain steady immune control.

NKG2D-based CARs enable the activation of innate immune recognition mechanisms. The constructs that utilize the full NKG2D receptor ensure a wide range of action versus phenotypically heterogenous myeloma cells [175,176]. In early-phase clinical trials, the efficacy of NKG2D-CAR T cells has been demonstrated in patients with acute myeloid leukemia, myelodysplastic syndrome, and relapsing/refractory MM. The application of full NKG2D receptor-based autologous CAR T cells with the CD3ζ signaling domain has indicated good tolerability and clinical activity signs [179].

The development of improved constructs has been underway, which include co-stimulating modules, such as DAP10 and DAP12. This is aimed at the enhancement of metabolic activity, resistance to depletion, and the prolonged persistence of CAR cells in the immunosuppressive tumor microenvironment [2,78].

Thus, the system of NKG2D-NKG2D ligands is a versatile and selective target for MM cellular therapy. It has been particularly relevant in pronounced tumor heterogeneity and the augmentation of therapeutic resistance, ensuring the potential for the development of steady and effective treatment strategies.

### 3.9. CD19 as a Therapeutic Target for Cellular Immunotherapy

CD19 is a transmembrane glycoprotein from the immunoglobulin superfamily and is expressed primarily on the membrane of mature and immature B lymphocytes [180]. It plays a key role in the activation, proliferation, and survival of B cells, functioning as a co-receptor in the composition of the B-cell receptor (BCR) signaling complex, which makes it a critical element for immune response regulation [181,182].

Despite the fact that MM is falling within the tumors of plasma cells, some surveys describe subpopulations of myeloma cells with residual expression of CD19, the presence of which is associated with an unfavorable forecast, relapses, and resistance to therapy [39,183,184]. This justifies the interest for CD19 as an extra therapeutic target within the scope of multi-target approaches.

The antigen escape phenomenon remains one of the main problems. It is related to mutations, expression disorder, and alternative splicing of mRNA or disorder in the formation of disulfide bridges, which results in the loss of CD19 expression [185,186]. Nonetheless, in preclinical trials CD19-specific CAR T cells have demonstrated pronounced activity versus CD19-positive subclones of myeloma cells, both in the mono-specific format and as parts of tandem constructs jointly with anti-BCMA CARs [187,188]. Moreover, such cell products have revealed selectivity versus the tumor cells with minimal impact on the normal subpopulations of immune cells [2].

Dual- (CD19 and BCMA) and poly-specific targeting platforms have been actively developed in recent years, which enables us to effectively overcome intraclonal heterogeneity, prevent antigen escape, and minimize the MRD growth rate [189,190,191,192]. Likewise the strategies that provide for successive or combined injection of CD19- and BCMA-specific CAR products have been emerging, especially to patients after deep tumor reduction in BCMA-targeted therapy, where resistant CD19-positive clones survive [193,194,195].

Based on the accumulated preclinical and clinical data, CD19 has been viewed as a promising target for application in multi-target cellular immunotherapy in MM patients, notably in the occurrence of relapses and resistance to BCMA-targeted treatment [138].

### 3.10. Transmembrane Activator and CAML Interactor (TACI, TNFRSF13B)

TACI is a transmembrane receptor from the tumor necrosis factor (TNF) receptor superfamily and plays a key role in the regulation of plasma cell survival, differentiation, and homeostasis. In physiological conditions, TACI expression is primarily observed on activated B lymphocytes and plasma cells, where it performs immunoregulatory functions via its interaction with BAFF (B-cell activating factor) and APRIL (a proliferation-inducing ligand) [196]. The binding of these ligands with TACI activates the intracellular signaling cascades that regulate the production of antibodies, maintaining the survival of long-lived plasma cells and B-cell homeostasis.

TACI is often co-expressed with BCMA on the surface of malignant plasma cells in MM. As indicated by the trials, the co-expression of these molecules is observed in most of the myeloma clones, whereby TACI expression may be higher and more stable in the later stages of the disease [129,138]. These features acquire special significance against the background of a renowned escape mechanisms via BCMA loss, since it makes TACI an attractive standby therapeutic target [197].

Surveys on TACI-specific CAR T cells confirm their high efficacy versus TACI-positive MM cells in in vitro and in vivo systems [198]. These cells demonstrate quick expansion, steady persistence, pronounced tumor eradication, and a possibility for disease control during heterogenous expression of targets [199]. Thus, toxicity for untransformed cells is revealed at the minimal level.

This is the reason for the active adoption of bi-specific CAR constructs that simultaneously target BCMA and TACI [197,200]. Such an approach minimizes the risk of relapse caused by antigen escape and ensures a wide coverage of tumor subclones. State-of-the-art developments include the creation of fully humane CAR constructs and armored CAR T cells with enhanced signaling activity that are capable of overcoming the immunosuppressive impact of the tumor environment.

Hence, TACI inclusion as a target for CAR T therapy in MM is a promising field, corroborated by molecular logic and clinical relevance. The combined targeting of BCMA and TACI opens up opportunities for enhancing the depth and duration of remissions, namely in patients with a high relapse risk and heterogenous antigen expression.

### 3.11. A Proliferation-Inducing Ligand (APRIL)

APRIL is a part of the TNF family and appears as a natural ligand for BCMA and TACI receptors [201]. The interaction of APRIL with these receptors is fundamental for maintaining the viability, differentiation, and homeostasis of plasma cells, which is critical for their long-term functioning [202].

Since APRIL shows high affinity to BCMA and TACI receptors, CAR constructs have been developed, in which the modified APRIL is used as an antigen-binding domain. This architecture is beneficial for the capability of simultaneously targeting two targets with no need to apply complex, in terms of engineering, bi-specific antibodies. This simplifies the structure of CAR platforms and increases their versatility and flexibility [203].

As indicated by preclinical trials, APRIL-CAR T cells effectively destroy tumor clones that express BCMA, TACI, or both the markers, which ensures a wider coverage of tumor heterogeneity [204]. Such dual targeting mitigates the risk of antigen escape—a key restriction for conventional CAR T platforms—and helps achieve deep and steady remission [205].

Additional optimizations for the spatial configuration of the APRIL domain in the CAR construct, as well as enhancing signaling modules (for instance, the inclusion of the co-stimulating CD28 or 4-1BB domain), have enabled a substantial increase in persistence and functional activity of effector cells, especially in the immunosuppressive microenvironment, which is typical for MM [206].

Hence, APRIL-targeted CAR platforms form one more promising field for cellular therapy of multiple myeloma, ensuring expansion of the range, a decrease in the relapse likelihood, and enhancement of the overall treatment efficacy.

### 3.12. Tumor Necrosis Factor Receptor 2 (TNFR2, TNFRSF1B)

TNFR2 is a transmembrane protein of the TNF receptor family. Unlike TNFR1, its expression is restricted within a few cell types, such as regulatory T cells (Tregs); myeloid-derived suppressor cells; the endothelium; tumor cells with diverse histogenesis; including MM cells; and microenvironment cells [207,208]. Such selective expression underpins the key role of TNFR2 in the formation of the immunosuppressive microenvironment and maintaining the functional activity of Tregs, which contribute to tumor resistance to immune surveillance [209].

The TNFR2 construct includes four extracellular cysteine-rich domains (CRDs), a transmembrane section, and an intracellular domain, and it lacks a death domain, which differentiates it from TNFR1. This enables the activation of alternative signaling cascades, including NF-κB and PI3K/AKT, which contribute to survival, proliferation, and immunoregulation [210]. The presence of several CRDs makes TNFR2 a promising target for high-affinity CAR constructs.

TNFR2-specific CAR T cells demonstrate the ability to suppress the growth of TNFR2-positive tumors via the remodeling of the tumor microenvironment and the suppression of Tregs [211,212]. Moreover, bi-specific CAR platforms that simultaneously recognize TNFR2 and other antigens, such as BCMA or GPRC5D, which boost selectivity and mitigate the risk of antigen escape, have been developed [213].

The creation of armored CAR products with resistance to immunosuppressive factors is aimed at the enhanced functional stability of cells and proliferative potential in unfavorable tumor environments [214]. The promising outlook of such an approach is confirmed by results from early-phase clinical trials, which evaluate the safety of TNFR2-CAR T in patients with relapsing tumors, including MM [215]. Moreover, TNFR2-specific CAR NK cells capable of effectively infiltrating the tumor and destroying the TNFR2-positive cells by secretion of interferon-γ and granzymes are undergoing active development [208,212].

Despite predominantly tumor-associated expression, TNFR2 is represented on the surface membrane of Tregs and is engaged in the regulation of their activity. This fact requires careful attention—excessive suppression of the subpopulation may lead to the occurrence of autoimmune diseases. Therefore, we recognize the use of platforms with regulated activation and safety switch systems that ensure the controllable action of CAR cells [214].

In general, TNFR2 is a promising cellular immunotherapy target that is focused on overcoming immunosuppression, enhancing the anti-tumor response, and expanding the opportunities for personalized treatment of resistant multiple myeloma forms.

### 3.13. CD44v6 as a Target for Cellular Therapy of Multiple Myeloma

CD44v6 is a variant isoform of the CD44 transmembrane glycoprotein, and it is derived from alternative splicing with the inclusion of the v6 exon into the molecule’s extracellular domain [216]. This modification drastically alters the spatial structure of the extracellular section, enhancing its interaction with hyaluronic acid, metalloproteases, VEGF (vascular endothelial growth factor), and HGF (hepatocyte growth factor), which promotes the invasiveness and survival of tumor cells [217,218].

CD44v6 overexpression is typical for aggressive MM forms and is associated with disease progression. CD44v6 expression is observed in 17% of patients in early MM stages and reaches 43% in the case of plasma cell leukemia [219], whereby it is often linked with 13q14 deletion—an unfavorable prognostic marker [220].

CD44v6 is deemed as a promising target for immunotherapy owing to its restricted expression in normal tissues and its high abundance on tumor cells. However, its presence on keratinocytes and the epithelium of mucous membranes creates the risk for the occurrence of on-target, off-tumor toxicity, which requires the development of safe engineering solutions [221].

This is the purpose for applying low-affinity CAR constructs that are capable of differentiating cells with high and low Cd44v6 expression, as well as safety switch systems for urgent inhibition of effector cells if complications develop [222]. Particularly interesting are CAR-NK platforms, which demonstrate innate anti-tumor activity and a reduced rate of CRS incidence as compared with CAR T [24,223].

Preclinical trials have confirmed the high efficacy of both CD44v6-CAR T, and CAR NK cells versus CD44v6-positive tumors, without pronounced damage to normal tissues [224,225]. The first clinical trials (phases I/IIa) of CD44v6-targeted CAR products in patients with refractory hematological solid tumors, including MM, have demonstrated controllable safety profiles and preliminary efficacy [226,227]. CD44v6 as a target is a part of poly- and bi-specific CAR platforms and is used to overcome antigen escape and intratumoral heterogeneity [228].

Hence, CD44v6 is a prominent target for cellular therapy of multiple myeloma with high therapeutic potential, provided that meticulous engineering optimization of safety is conducted.

### 3.14. Alternative Targets for CAR Cell Therapy in Multiple Myeloma

Further, concerning the best known targets for CAR therapy in MM, preclinical trials on cellular products that target alternative molecules possessing therapeutic potential have been conducted [129]. These include CD56, an activated form of integrin β7 [229], CD123, and the tumor-associated antigen Lewis-Y [230]. The choice of these targets is due to restricted expression in normal tissues and engagement in MM pathogenesis, which make them attractive for the development of selective CAR cell constructs [124,231].

CD56, also known as neural cell adhesion molecule (NCAM), is expressed in more than 70% of MM patients and plays a crucial role in the migration and adhesion of tumor cells in the bone marrow [232,233]. The absence of CD56 expression is associated with a more aggressive course of multiple myeloma and a trend for increasing the quantity of plasma cells in the peripheral blood [234]. In normal tissues CD56 expression is primarily restricted to NK cells, which required a comprehensive safety evaluation in the development of CD56-specific CAR cells. Nevertheless, preclinical data have shown tangible anti-tumor activity of CAR T-cells targeted against this antigen [163].

Integrin β7, especially in its activated form, has been viewed as a promising therapeutic target, since it is primarily expressed on MM cells [235]. The use of the MMG49 monoclonal antibody, which selectively recognizes the activated form of integrin β7, has enabled us to engineer CAR constructs with high antigen specificity and mitigated the risk of off-tumor toxicity [235]. Preclinical trials have demonstrated the capability of such CAR cells to selectively lyse multiple myeloma cells while not affecting healthy hematopoietic cells [122].

CD123 (α-subunit of the interleukin-3 receptor) is an antigen which is expressed on tumor stem cells and plasmacytoid dendritic cells, and it is engaged in the formation of the immunosuppressive microenvironment in multiple myeloma [236,237]. Despite being actively investigated as a therapeutic target in acute myeloid leukemia, CD123 potential in therapy for multiple myeloma remains understudied [238,239]. Restricted CD123 expression in normal tissues opens up the prospects for its use in immunotherapy, but additional surveys are needed in light of possible hematological toxicity.

Lewis-Y is a tumor-associated carbohydrate antigen, and it is expressed on the surface of cells in various malignant neoplasms, including MM [240]. In preclinical trials Lewis-Y-specific CAR T cells have demonstrated pronounced anti-tumor activity [240,241]. A phase I clinical trial has shown cases of temporary remission and disease stabilization for a period of up to 23 months [242]. Moreover, the infiltration and prolonged persistence of CAR T cells (up to 10 months) has been confirmed, which is deemed a crucial factor for the efficacy of the therapy [230].

Despite the encouraging preclinical and limited clinical results, the scope of data accumulated on alternative targets yields BCMA-targeted strategies. The matters of toxicity, antigen escape, and therapeutic response stability have remained unsolved [129]. In this connection, the development of multi-antigen CAR constructs and engineering platforms with controllable activation of effector cells (for instance, systems like the AND gate), as well as the adoption of safety systems for the quick and controllable suppression of cytotoxic activity should any undesired effects occur, has been very relevant [2,243,244].

Hence, CD56, integrin β7, CD123, and Lewis-Y appear as biologically substantiated targets for further development of CAR cellular therapy for multiple myeloma. Their integration into clinical practice will require additional surveys with a focus on the safety, specificity, and long-term efficacy of the therapeutic response [122].

## 4. Conclusions

MM is characterized by a high relapse incidence and the formation of multiple drug resistance mechanisms. No radical recovery from the disease has been obtained so far, which necessitates the introduction of game-changing therapeutic approaches.

CAR cell therapy is an area with the most breakthroughs in treatment for relapsing and refractory forms of MM. Its efficacy is defined by multiple factors, with the rational choice of tumor target being the key one. The presented review has analyzed data on the biological properties, level, and stability of the expression, selectivity, and clinical development stages of target molecules, such as BCMA, GPRC5D, FcRH5, SLAMF7, CD38, CD138, TACI, APRIL, CD19, CD44v6, TNFR2, CD70, NKG2D ligands, etc.

BCMA is the most verified and clinically proven target for CAR cell therapy in MM. However, despite the high frequency of a therapeutic response, the use of this target shows some restrictions, including the development of antigen-negative relapses and the presence of the soluble BCMA form (sBCMA), which decreases the efficacy of cellular therapy.

The most promising targets in the advanced stage of clinical development include GPRC5D and FcRH5, which demonstrate high specificity to myeloma cells, the favorable safety profile, and efficacy in patients who had earlier received BCMA-specific therapy. The development of bi-specific CAR products that simultaneously target BCMA and GPRC5D/FcRH5 has shown encouraging results in overcoming antigen escape and increasing the stability of a clinical response.

Second-line targets, such as SLAMF7, CD38, CD138, TACI, and APRIL, possess biological validity, high expression on myeloma cells, and proven therapeutic potential in preclinical and early clinical trials. Meanwhile, the use of some targets requires particular care due to the risk of on-target, off-tumor toxicity or involvement in the regulation of immune homeostasis. These include CD44v6, TNFR2, CD70, and CD19, which are also expressed on normal epithelial or immune cells. CAR logical activation strategies (AND/NOT gates), low-affinity receptors, and controllable inactivation systems (safety switches) are used to increase therapeutic effect selectivity and controllability with respect to these targets.

Alternative targets, such as CD56, the activated form of integrin β7, CD123, and Lewis-Y, are primarily in the preclinical validation stage.

The choice of a tumor target in multiple myeloma is determined by a set of parameters. The optimization of antigen targeting via multi-specific CAR constructs, logically programmed platforms, and controllable safety mechanisms is a key field in the development of more effective, steady, and safe forms of cellular therapy in MM.

## Figures and Tables

**Figure 1 ijms-26-06051-f001:**
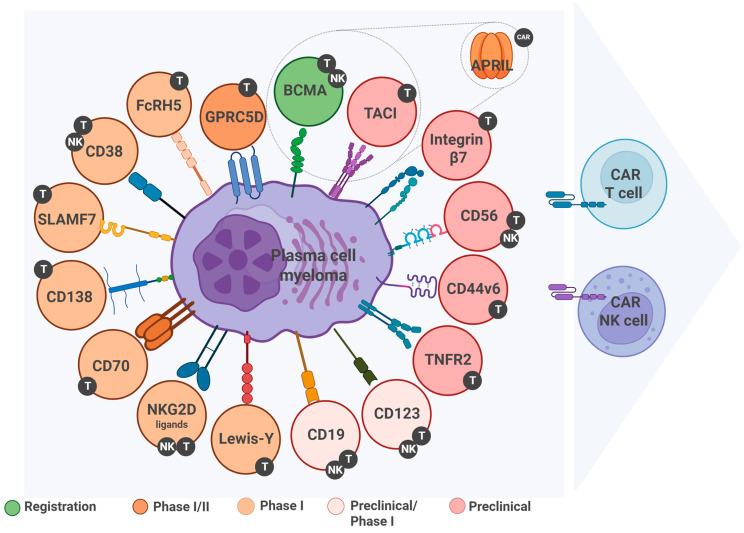
Targets in multiple myeloma and stages of immunotherapeutic development. This schematic presents targets investigated in CAR-T and CAR-NK cell therapies for MM, categorized by their clinical development stage. Green indicates validated membrane-bound targets with approved CAR-T products (e.g., BCMA). Orange corresponds to targets in Phase I–II trials; light orange to those in Phase I; light pink to targets at the preclinical/Phase I interface; and pink to preclinical-stage targets. The color of each target represents the most advanced development stage achieved for either CAR-T or CAR-NK platforms. Circular markers with “T” or “NK” next to each target indicate whether the target is being explored in CAR-T cells, CAR-NK cells, or both (dual-labeled if applicable), based on available clinical or preclinical evidence. All targets are membrane-bound surface proteins suitable for direct CAR recognition, except APRIL, which is labeled with a “CAR” circle to indicate that it is not a target, but a ligand incorporated into CAR-T constructs as a binding domain, enabling simultaneous of BCMA and TACI. NKG2D ligands (e.g., MICA, MICB, ULBP family) are stress-induced molecules transiently expressed on the tumor cell surface and are targeted using CAR constructs that incorporate the native NKG2D receptor. This figure illustrates the diversity of immunotherapeutic targets in MM and highlights the rationale for multi-target strategies to overcome tumor heterogeneity and antigen escape. Figure created in https://BioRender.com (accessed on 15 June 2025).

**Table 1 ijms-26-06051-t001:** Parameters of tumor targets for CAR therapy.

Target	Expression	Expression in Normal Tissues	Risk of Off-Tumor Toxicity	Development Stage	Clinical Trial ID (If Available)	References
BCMA	High and stable	Plasma cells	Low	Registration (ide-cel, cilta-cel)	NCT03361748 and NCT04162210	[80,81]
GPRC5D	High	Squamous epithelium	Low	Phase I/II	NCT04555551	[82]
FcRH5	Moderately high	Few	Low	Case Report	ChiCTR2000041025 *	[83]
CD38	High	Immune and epith. cells	Average	Phase I	NCT04351022	[84]
SLAMF7	Moderate	NK, DC, and T cells	Average	Phase I	NCT03958656	[85]
CD138	High	Epithelium	Average	Phase I	NCT04430530	[86]
CD70	Moderate	Activ. immune cells	Average	Phase I	NCT02830724	[87]
NKG2D ligands	Heterogenous	Minimal	Low	Phase I	NCT04623944	[88]
CD19	Residual (subclones)	B cells	Average	Preclinical/Phase I	NCT02135406	[89]
TACI	Variable	B cells	Low	Preclinical		
APRIL	No data available	No data available	No data available	Preclinical		
TNFR2	Moderate	Tregs and stroma	Average/high	Preclinical		
CD44v6	Variable	Keratinocytes	High	Preclinical		
Alternative targets for CAR cell therapy in multiple myeloma						
CD56	70% of cases	NK cells	Average	Preclinical		
Integrin β7	Prevalently	Minimal	Low	Preclinical		
CD123	Low	Plasm. DC	Average	Preclinical/Phase I (AML)	NCT04318678, NCT02159495, NCT02623582, and NCT04599543 (only AML, not MM)	[90,91,92,93]
Lewis-Y	≈50%	Low	Average	Phase I	NCT01716364	[94]

*—www.chictr.org.cn as ChiCTR2000041025.

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
