# Peer review of "Targets for CAR Therapy in Multiple Myeloma"

_ijms, 2025, doi:10.3390/ijms26136051_

Round 1
Reviewer 1 Report
Comments and Suggestions for Authors
nothing to comment on apart from reporting some extra data on Response rates and/or Overall survival in clinical trials with the new agents. Same stands for the in vitro data with the MM lines
============================================
Further comments
There’s a plethora of targets being tested as putative therapeutic avenues. The manuscript is a catalogue of these targets. It is a thorough presentation listing everything from clinical trials to preclinical data. I just recommended the addition of OS/response data for a more objective view of the literature.Comments on the Quality of English Language
few mistakes (dur instead of Due, oftener !!! allogenous !!!)
Author Response
Reviewer I
Comments and Suggestions for Authors
nothing to comment on apart from reporting some extra data on Response rates and/or Overall survival in clinical trials with the new agents. Same stands for the in vitro data with the MM lines
"There’s a plethora of targets being tested as putative therapeutic
avenues. The manuscript is a catalogue of these targets. It is a
thorough presentation listing everything from clinical trials to
preclinical data. I just recommended the addition of OS/response data
for a more objective view of the literature."
Comments on the Quality of English Language
few mistakes (dur instead of Due, oftener !!! allogenous !!!)
Dear Reviewer I,
Thank you for reviewing our manuscript, also for your valuable comments and suggestions for its improvement.
We revised our manuscript according to your suggestions:
- Table 1 was supplied with clinical registration numbers and links to it inserted into the Reference List. It could be mentioned though that not so many completed clinical trials have been described so far. The obtainable data from literature, both on preclinical and clinical trials, are included and discussed in our review.
- We submitted the revised manuscript to MDPI Author Services for editing its English grammar.
Kind regards,
Olga Bezborodova and co-authors

Reviewer 2 Report
Comments and Suggestions for Authors
Review Comments
This is a meaningful study focuses on the targets for CAR therapy in multiple myeloma, which will arouse readers' interest to a certain extent. In my view, this article can be reconsidered for the publication in International Journal of Mechanical Sciences after major revisions, and the suggestions are mentioned below:
- The overview content of this review article should be supplemented at the end of the Introduction section.
- The segmentation of Section 2 should be reconsidered since there is a lot of content relevant to the background of this section, while there is only one subsection (Section 2.1).
- The description of OR gates should be supplemented between Line 164 to 172.
- The references or registry IDs relevant to each parameter of tumor targets for CAR therapy should be supplemented in Table 1.
- The citation numbers of literature should be arranged from small to large in this manuscript (e.g. the “[112, 111]” in Line 342, “[133, 84]” in Line 402, etc. should all be corrected).
- No figures can be found in this manuscript, and at least one figure should be added in this manuscript to summarize the content of the whole review article, or the specific section.
- The Conclusion section should be further simplified.
- DOI should be applied in Referfence section instead of Website URL.
- Some minor errors: 1) The phrase “Dur to” in Line 380 should be changed to “Due to”; 2) the citation numbers of different literature should be separated by commas instead of semicolons (e.g., the “[138; 139]” in Line 416, “[2; 102]” in Line 688).

Author Response
This is a meaningful study focuses on the targets for CAR therapy in multiple myeloma, which will arouse readers' interest to a certain extent. In my view, this article can be reconsidered for the publication in International Journal of Mechanical Sciences after major revisions, and the suggestions are mentioned below:
- The overview content of this review article should be supplemented at the end of the Introduction section.
- The segmentation of Section 2 should be reconsidered since there is a lot of content relevant to the background of this section, while there is only one subsection (Section 2.1).
- The description of OR gates should be supplemented between Line 164 to 172.
- The references or registry IDs relevant to each parameter of tumor targets for CAR therapy should be supplemented in Table 1.
- The citation numbers of literature should be arranged from small to large in this manuscript (e.g. the “[112, 111]” in Line 342, “[133, 84]” in Line 402, etc. should all be corrected).
- No figures can be found in this manuscript, and at least one figure should be added in this manuscript to summarize the content of the whole review article, or the specific section.
- The Conclusion section should be further simplified.
- DOI should be applied in Referfence section instead of Website URL.
- Some minor errors: 1) The phrase “Dur to” in Line 380 should be changed to “Due to”; 2) the citation numbers of different literature should be separated by commas instead of semicolons (e.g., the “[138; 139]” in Line 416, “[2; 102]” in Line 688).
Dear Reviewer II,
Thank you for reviewing our manuscript, also for your valuable comments and suggestions for its improvement.
We revised our manuscript according to your suggestions:
- The overview content of our review was added and placed at the end of the Introduction Section
- The segmentation of Section 2 was reconsidered and we divided it into subsections according to the content
- The description of OR-gates was included into the appropriate section
- Table 1 was supplied with clinical registration numbers and links to it inserted into the Reference List
- The numbers of the cited literature were arranged properly
- The Figure summarizing the review content was added to the manuscript
- Taking into account the number of the described targets and a long Reference List the Conclusion Section seems to be quite adequate. It shortly summarizes the review and in our opinion could be left unaltered
- DOI were inserted into the Reference section instead of the Website URL
- We submitted the revised manuscript to MDPI Author Services for editing its English grammar.
Kind regards,
Olga Bezborodova and co-authors

Reviewer 3 Report
Comments and Suggestions for Authors
This is a quite comprehensive review, summarized and discussed the most prevalent CAR targets for treating MM in detail. However, I have the following concerns that may need to be addressed before publishing:
- Line 64, change the sentence to“the typical extracellular domain” since some may use a portion of the natural receptors as theextracellular domain, i.e. NKG2D.
- Line 70, better to add “the most well-known cellular products…” as many other immune cells nowadays startedto be used in cancer immunotherapy.
- Line 73 and 80, please emphasize that this is a typical structure, not for all circumstances.
- Line 92, NKG2D recognizes NKG2DLigands (in total eight), MICA and MICB are only two of them.
- Line 106, universal CAR-T also have the potential to be used as “off-the-shelf”
- Please keep all the terms consistent, CAR constructs can be typically used on both T cells and NK cells,
- The section 2 is more like a general intruduction of CAR constructs and CAR immune cells. I would suugest to shorten it or try to link it more to the topic of the disease MM.
- I would like to suggest the authors to add the clinical registration number in the Table I for readers to better access to the clinical trial data (if applicable).
- Line 312, this paragraph seems to be a general problem not specific for BCMA.
- Line 380, typo “Dur”, “offener” is not a proper english word.
- Check the typo of “CD38” (Line 417 and etc.).
- The author may consider state the logic of arrangment of the different targets of MM (from more developed to less developed). The subtitle of each of the target is also needed to be standardized.
- Add the references for Line 684.
Comments on the Quality of English Language
Please read the article again to check the grammar mistakes carefully, especially the ones with long sentence structure, which may confuse readers significantly.
Author Response
Reviewer III
Comments and Suggestions for Authors
This is a quite comprehensive review, summarized and discussed the most prevalent CAR targets for treating MM in detail. However, I have the following concerns that may need to be addressed before publishing:
- Line 64, change the sentence to“the typical extracellular domain” since some may use a portion of the natural receptors as theextracellular domain, i.e. NKG2D.
- Line 70, better to add “the most well-known cellular products…” as many other immune cells nowadays startedto be used in cancer immunotherapy.
- Line 73 and 80, please emphasize that this is a typical structure, not for all circumstances.
- Line 92, NKG2D recognizes NKG2DLigands (in total eight), MICA and MICB are only two of them.
- Line 106, universal CAR-T also have the potential to be used as “off-the-shelf”
- Please keep all the terms consistent, CAR constructs can be typically used on both T cells and NK cells,
- The section 2 is more like a general intruduction of CAR constructs and CAR immune cells. I would suugest to shorten it or try to link it more to the topic of the disease MM.
- I would like to suggest the authors to add the clinical registration number in the Table I for readers to better access to the clinical trial data (if applicable).
- Line 312, this paragraph seems to be a general problem not specific for BCMA.
- Line 380, typo “Dur”, “offener” is not a proper english word.
- Check the typo of “CD38” (Line 417 and etc.).
- The author may consider state the logic of arrangment of the different targets of MM (from more developed to less developed). The subtitle of each of the target is also needed to be standardized.
- Add the references for Line 684.
Comments on the Quality of English Language
Please read the article again to check the grammar mistakes carefully, especially the ones with long sentence structure, which may confuse readers significantly.
Dear Reviewer III,
Thank you for reviewing our manuscript, also for your valuable comments and suggestions for its improvement.
We revised our manuscript according to your suggestions:
1-6. The text of the manuscript was revised in accordance with your comments
- The segmentation of Section 2 was reconsidered and we divided it into subsections according to the content
- Table 1 was supplied with clinical registration numbers and with references which were also inserted into the Reference List
- We agree with you that solubility is a common problem for all discussed targets. But since it fully applies to BCMA we do not think it is a mistake and as such could be kept in this section.
- We submitted the revised manuscript to MDPI Author Services for editing its English grammar.
- The typo of “CD38” was checked and corrected throughout the text.
- Taking into account your suggestion we have slightly changed the logic of description of the different targets. In the revised manuscript the targets are described according to the level of their previous study. The same order is applied to Table 1.
- The Reference #188 is added to the statement in Line 684.
- We submitted the revised manuscript to MDPI Author Services for editing its English grammar.
Kind regards,
Olga Bezborodova and co-authors

Round 2
Reviewer 2 Report
Comments and Suggestions for Authors
Review Comment
1. This review article can be published in International Journal of Molecular Sciences after further simplify the Conclusion section because the present form of this section is still not concise and compact enough.
Author Response
This review article can be published in International Journal of Molecular Sciences after further simplify the Conclusion section because the present form of this section is still not concise and compact enough.
Dear Reviewer,
Thank you for reviewing our manuscript, also for your valuable comment for its improvement.
We revised our manuscript according to your suggestion:
The correction of the manuscript consists of simplifying the “Conclusion” section and making it more compact.
This is the new version of the “Conclusion” section:
- Conclusion
MM is characterized by high relapse incidence and formation of multiple drug resistance mechanisms. No radical recovery from the disease has been obtained so far, which necessitates introduction of game-changing therapeutic approaches.
CAR cell therapy is a most breakthrough area in treatment for relapsing and refractory forms of MM. Its efficacy is defined by multiple factors, rational choice of tumor target being the key one. The presented review has analyzed data on biological properties, level and stability of expression, selectivity, and clinical development stages of target molecules, such as BCMA, GPRC5D, FcRH5, SLAMF7, CD38, CD138, TACI, APRIL, CD19, CD44v6, TNFR2, CD70, NKG2D ligands, etc.
BCMA is the most verified and clinically proven target for CAR cell therapy in MM. However, despite the high frequency of therapeutic response, the use of this target shows some restrictions, including development of antigen-negative relapses and presence of the soluble BCMA form (sBCMA), which decreases the efficacy of cellular therapy.
The most promising targets in the advanced stage of clinical development include GPRC5D and FcRH5, which demonstrate high specificity to myeloma cells, the favorable safety profile and efficacy in patients who had earlier received BCMA-specific therapy. Development of bispecific CAR products, which simultaneously target BCMA and GPRC5D/FcRH5, has shown encouraging results in overcoming antigen escape and increasing stability of clinical response.
Second-line targets, such as SLAMF7, CD38, CD138, TACI and APRIL, possess biological validity, high expression on myeloma cells and proven therapeutic potential in preclinical and early clinical trials. Meanwhile, the use of some targets requires particular care, due to the risk of on-target, off-tumor toxicity or involvement in regulation of immune homeostasis. These include CD44v6, TNFR2, CD70 and CD19, which also express on normal epithelial or immune cells. CAR logical activation strategies (AND/NOT gates), low-affinity receptors and controllable inactivation systems (safety switches) are used to increase therapeutic effect selectivity and controllability in respect of these targets.
Alternative targets, such as CD56, activated form of integrin β7, CD123 and Lewis-Y, are primarily in the preclinical validation stage.
The choice of a tumor target in multiple myeloma is determined by a set of parameters. Optimization of antigen targeting via multi-specific CAR constructs, logically programmed platforms and controllable safety mechanisms is a key field in development of more effective, steady and safe forms of cellular therapy in MM.
